# HSPA2 Chaperone Contributes to the Maintenance of Epithelial Phenotype of Human Bronchial Epithelial Cells but Has Non-Essential Role in Supporting Malignant Features of Non-Small Cell Lung Carcinoma, MCF7, and HeLa Cancer Cells

**DOI:** 10.3390/cancers12102749

**Published:** 2020-09-24

**Authors:** Damian Robert Sojka, Agnieszka Gogler-Pigłowska, Katarzyna Klarzyńska, Marta Klimczak, Alicja Zylicz, Magdalena Głowala-Kosińska, Zdzisław Krawczyk, Dorota Scieglinska

**Affiliations:** 1Center for Translational Research and Molecular Biology of Cancer, Maria Sklodowska-Curie National Research Institute of Oncology, Gliwice Branch, 44-102 Gliwice, Poland; damian.sojka@io.gliwice.pl (D.R.S.); agnieszka.gogler-piglowska@io.gliwice.pl (A.G.-P.); katarzyna.klarzynska@gmail.com (K.K.); zdzislaw.krawczyk@io.gliwice.pl (Z.K.); 2International Institute of Molecular and Cell Biology, 02-109 Warsaw, Poland; m.klimczak01@gmail.com (M.K.); azylicz@iimcb.gov.pl (A.Z.); 3Department of Bone Marrow Transplantation and Oncohematology, Maria Sklodowska-Curie National Research Institute of Oncology, Gliwice Branch, 44-102 Gliwice, Poland; magdalena.glowala-kosinska@io.gliwice.pl

**Keywords:** HSPA2, HSP70, heat shock protein, bronchial epithelial cells, non-small cell lung carcinoma, breast cancer, cervical cancer, malignant phenotype, cell growth, adhesion

## Abstract

**Simple Summary:**

Heat shock proteins A (HSPA) are molecular chaperones that play a central role in the cellular protein quality control. One of them is HSPA2 which recently was indicated as a novel cancer-related protein due to its elevated expression in various tumors and reported prognostic significance. Several previous in vitro studies have shown significant role of HSPA2 in supporting cancer cells growth and invasiveness. Our data presented in this article contradict the current belief of the essential role of HSPA2 chaperone and show that HSPA2 is not crucial for maintenance of the malignant phenotype of lung, breast, and cervical cancer cells. Instead, we revealed HSPA2’s role in supporting clonogenic potential and adhesive ability of bronchial epithelial cells. Therefore, further research should concentrate on elucidating HSPA2 roles in epithelial cells.

**Abstract:**

Heat Shock Protein A2 (HSPA2) is a member of the HSPA (HSP70) chaperone family and has a critical role for male fertility. HSPA2 is present in a number of somatic organs. Limited evidence suggests that HSPA2 may be involved in regulating epithelial cell differentiation. HSPA2 also emerged as a cancer-related chaperone; however, no consensus on its functional significance has been reached so far. In this study, we compared the phenotypic effects of HSPA2 deficit in non-transformed human bronchial epithelial cells (HBEC), and in lung, breast, and cervical cancer cells. We used various techniques to inhibit the *HSPA2* gene expression in order to examine the impact of HSPA2 deficiency on cell growth, migration, adhesion, and invasion. Our results show that HBEC but not cancer cells are sensitive to HSPA2 deficit. HSPA2 knockdown in HBEC cells impaired their clone-forming ability and adhesiveness. Thus, our results indicate that epithelial cells can rely on a specific activity of HSPA2, but such dependence can be lost in epithelial cells that have undergone malignant transformation.

## 1. Introduction

Members of the multigene HSPA (HSP70) family encode molecular chaperone proteins approximately 70 kDa in size that create a machinery that has a central role in the cellular protein quality control. The HSPA family comprises some of the most evolutionary conserved proteins [1]. The human genome contains 13 protein-coding HSPA genes that are expressed either constitutively and/or can be upregulated/induced under conditions of proteotoxic stress caused by environmental or pathophysiological stimuli [2,3]. Some of the HSPA paralogs are expressed in a cell- or tissue-type specific manner or during specific developmental processes. In terms of intracellular distribution, five HSPA paralogs showing the highest degree of sequence homology, namely HSPA1, HSPA2, HSPA6, HSPA8, and HSPA1L shuttle between cytoplasm and nuclei upon physiologic and stressful conditions, respectively. Some of the HSPA family members localize in other cellular compartments; i.e., HSPA9 is present in mitochondria, and HSPA5 is present in endoplasmic reticulum [2,4]. Whether the functions of cytoplasmic/nuclear HSPA overlap or differ has not yet been sufficiently recognized [5]. Although it is generally accepted that their functionality is interchangeable, there is important evidence that particular HSPA paralogs may interact differently with the specific client protein(s) and thus have distinct functional significance [6,7,8]. 

The pattern of HSPAs expression may change during various pathophysiological states, including carcinogenesis. The vast majority of human tumors contain highly elevated levels of HSPAs, in particular the stress-inducible paralogs (e.g., HSPA1). It is believed that the direct involvement of HSPAs provides a selective advantage to malignant cells by suppressing death pathways, bypassing cellular senescence program, interfering with tumor immunity, promoting angiogenesis, and supporting metastasis [9,10]. The contribution of HSPAs to most of the cancer hallmarks represents the phenomenon called “non-oncogenic addiction” implicating the dependence of cancer cells on HSPA’s activity [11]. One of the highly important but still debatable issues is whether and to what extent the malignant phenotype is dependent on a particular HSPA paralog or relies on HSPAs as a group of functionally interchangeable chaperones. The first possibility is supported by data suggesting that the stress-inducible HSPA1 protein, which is produced constitutively at high levels in various cancers, supports viability and/or contributes to the high resistance of cancer cells to the anticancer treatment [12,13,14]. HSPA1 may also play a pivotal role in cancer development [15]. On the contrary, other reports show that only the simultaneous inhibition of several HSPA paralogs (including HSPA8), but not a paralog-selective reduction in the level of any of them, had anti-proliferative or cytotoxic effects in cancer cells [16,17,18]. 

HSPA2 was initially considered as testis-specific chaperone crucial for male germ cell development and function [4,19]. Recently, it has emerged as a novel cancer-related protein due to its elevated expression in various tumors and reported prognostic significance (for summary, we refer to [20]). However, HSPA2 is also expressed in a cell-type specific manner in a number of human somatic organs [21,22]. Specifically, high levels of HSPA2 were detected, among others, in the glial cells, as well as in the basal cells in the epidermis and in esophageal and bronchial epithelia, in goblet cells in the colon epithelium, and in the smooth muscle fibers of lamina muscularis mucosae [21]. 

HSPA2 is one of the least characterized HSPA paralogs, and the current understanding of its role(s) in somatic and cancer cells remains very limited. It was shown that HSPA2, similarly to HSPA1, was able to refold heat-denatured luciferase and prevent protein aggregation [7]. In a heterologous cellular system void of HSPA1 expression, HSPA2 protected hamster fibroblasts against heat-shock induced injury [23]. In turn, the overexpression of HSPA2 in the neuroglioma H4–4R0N cells increased levels of pathologically processed amyloid (b40, and b42) in the HEK293SW cells, and it also increased levels of total and phosphorylated tau protein [24]. An integrated, multi-level analysis of DNA, RNA, and protein data revealed HSPA2 as a top target and as a key driver in late-onset Alzheimer’s disease brain tissues [24]. The assessment of endogenous HSPA2 in a physiologically relevant context confirmed its functional importance in somatic cells. The *HSPA2* gene knockdown in epidermal keratinocytes revealed that its protein product was required to maintain their undifferentiated phenotype but not to provide protection against heat shock-induced toxicity [25]. 

Studies on cancer cells have not yet given conclusive results regarding the role of HSPA2 in cancer. The earliest study in cancer cells showed that the small interfering RNA (siRNA)-mediated partial knockdown of *HSPA2* significantly reduced growth and produced a distinct phenotype from that caused by the *HSPA1A/B* knockdown, thus pointing to a possible functional diversity between HSPA1 and HSPA2 [26]. The essential role of HSPA2 in supporting viability, motility, adhesiveness, and invasiveness was also revealed in studies performed on different cancer cell lines after transient shRNA-mediated knockdown of HSPA2 [27,28]. Conversely, using siRNA-mediated gene silencing, it was shown that neither HSPA2 nor HSPA1 were essential to cancer cells’ viability [17]. In our recent study, we demonstrated that the growth and proliferation of two NSCLC cell lines remained unaltered after the stable shRNA-mediated single knockdown of *HSPA2* or *HSPA1A/B*, as well as double *HSPA2*/*HSPA1A/B* knockdown [18].

A lack of consensus on the HSPA2 significance in cancer cells may suggest that both the regulation of the *HSPA2* expression and contribution of the encoded chaperone to the biology of normal versus tumor cells are complex. One may suspect several scenarios for HSPA2 significance in cancer and corresponding non-tumorigenic cells. First, HSPA2 may gain new crucial importance in cancer cells while being non-essential in corresponding non-transformed cells. Secondly, HSPA2 may have essential but different contributions to the phenotype of cancer and corresponding non-transformed cells. In the present work, we tried to address the above questions by analyzing the phenotypic effects of HSPA2 deficit on human bronchial epithelial cells (HBEC) and non-small cell lung carcinoma (NSCLC) cell lines. We also examined the effects of *HSPA2* knockdown on the malignant phenotype of selected breast and cervical cancer cell lines that have been previously identified as dependent on the HSPA2 protein. Our results show that HSPA2 contributes to HBEC phenotype, but its deficit has a negligible impact on the viability, growth, migration, invasion, and adhesion of cancer cells.

## 2. Results

### 2.1. HSPA2 Knockdown Reduces Colony-Forming Capacity of HBEC but Not NSCLC Cells, although It Influences neither Proliferation nor Metabolic Activity of HBEC and NSCLC Cells

We examined the effects of *HSPA2* deficiency on the phenotype of immortalized bronchial epithelial BEAS-2B cells, as a representative of HBEC cells, and four NSCLC cell lines that differed in HSPA2 protein levels. In NCI-H1299, NCI-H358, and NCI-H23 cell lines, the protein level of HSPA2 was high, while it was low in NCI-H520 cell line (Figure 1a). To address this question, three *HSPA2*-targeting short hairpin RNA (shRNA) sequences (shRNAG3, shRNA3, and shRNA4) and one control non-targeting shRNA sequence (shRNA-luc targeting firefly luciferase gene) were transduced (using lentiviral vectors) into HBEC and NSCLC cell lines. Figure 1b shows the effectiveness of HSPA2 knockdown in HBEC and NSCLC cells. 

Our first objective was to examine whether inhibition of the *HSPA2* gene expression has an impact on the proliferation, metabolic activity, and colony-forming ability of HBEC cells. The highest reduction in HSPA2 protein level was achieved in sh-4 cells transduced with shRNA4 sequence (Figure 1b and Figure 2a). The silencing was specific for the target gene, as the expression levels of other highly homologous HSPA paralogs (HSPA1 and HSPA8) remained unaffected in HSPA2-deficient cells (Figure 2a). Importantly, the protein levels of HSPA2 were similar in wild-type (wt) and the control sh-luc cells transduced with shRNA-luc sequence (Figure 2a). 

We found that the control and HSPA2-deficient cell lines grown up to 96 h show comparable metabolic activity (Figure 2b) and proliferation rates (Figure 2c). However, HSPA2 deficiency significantly decreased the ability of sh-4 cells to form colonies (Figure 2d). Thus, one can conclude that HSPA2 can support the clonogenic potential of HBEC cells.

Next, we examined whether HSPA2 deficiency would influence the growth characteristics of NSCLC cells. We established bronchioalveolar carcinoma NCI-H358 cell line and squamous cell carcinoma NCI-H520 cell line with a stable *HSPA2* knockdown (Figure 3a and Appendix A). Of note, in wild-type cells, the protein level of HSPA2 in relation to BEAS-2B was similar in NCI-H358 cells and highly reduced in NCI-H520 cells (Figure 1a)**.** The *HSPA2* gene silencing was highly effective in NCI-H358 and NCI-H520 cells transduced with shRNA3 or shRNA4 sequences (Figure 1b). Nevertheless, in both cell lines, neither metabolic activity and proliferation rate (Figure 3b,c and Appendix A), nor the potential to form colonies (Figure 3d and Appendix A) were dependent on the level of HSPA2 protein expression. Next, in order to verify our findings, we used retroviral vectors to generate *HSPA2* knockdown in NCI-H358 cells (Appendix A). We found that a high reduction of HSPA2 levels in modified cells had no impact on proliferation (Appendix A) and colony-forming ability (Appendix A). Altogether, these results were in line with our earlier observations that shRNA-mediated knockdown of *HSPA2* expression did not affect the proliferation, metabolic activity, and clone-forming ability of two cell lines originating from NSCLC, namely NCI-H23 (adenocarcinoma, arising from bronchial mucosal gland) and NCI-H1299 (derived from the metastatic tumor) [18].

In order to exclude the possibility that even a small amount of HSPA2 remaining after gene knockdown may be sufficient to support cell growth, we used the CRISPR/Cas9 (clustered regularly interspaced short palindromic repeats and CRISPR-associated protein 9) gene editing method. We generated *HSPA2* knockout in the NCI-H1299 cell line, which we had used previously in *HSPA2* knockdown study [18]. We established a knockout pool: a mixed population of edited CRISPR-A2 cells that contained around a 90% decreased level of HSPA2 (Figure 1b), as well as a control CRISPR-ctr pool. The editing was specific for the target gene, as the protein levels of highly homologous HSPA1 and HSPA8 proteins remained unaffected in *HSPA2* knockout cells (Figure 3e). We found that control and CRISPR-A2 cells showed comparable metabolic activity (Figure 3f), proliferation rate (Figure 3g), and clone-forming ability (Figure 3h). These results unequivocally confirmed that the role of HSPA2 in supporting the growth and proliferation of NSCLC cells is negligible.

### 2.2. HSPA2 Knockdown Reduces Adhesiveness of HBEC but Not NSCLC Cells, While Having No Impact on Their Motility

Our next goal was to examine whether HSPA2 deficit would affect the migratory and adhesive status of HBEC and NSCLC cells. Cell motility was examined using wound healing and Boyden chamber assays. We found that the rates of wound closure at 8 h after scratching confluent HBEC cell monolayers were similar for control (sh-luc) and HSPA2-deficient (sh-4) cells (Figure 4a). In addition, no differences between control and HSPA2-deficient cells were observed in the number of cells that migrated through the membrane in response to the extracellular chemical signals (fetal bovine serum (FBS) gradient, chemotaxis) (Figure 4b). These results clearly show that the migratory properties of HBEC were not changed by different levels of *HSPA2* expression. 

Next, we searched for a possible relation between the *HSPA2* gene expression and adhesiveness of HBEC cells using the colorimetric ECM Cell Adhesion Array Kit. We found that HBEC cells deficient in HSPA2 protein were less adhesive to fibronectin, laminin, and tenascin but not to collagens (I, II, and IV) or vitronectin (Figure 4c). These observations are in line with our previous findings showing that BEAS-2B cells deficient in HSPA2 were less adhesive to fibronectin but not to collagen IV [25]. Altogether, the results presented in Figure 4 allow us to conclude that HSPA2 in HBEC can play a role in modulating extracellular matrix (ECM) adhesiveness, but not motility. 

The effects of stable *HSPA2* knockdown in NCI-H1299, NCI-H358, and NCI-H23 cell lines on motility, chemotactic migration, and the ability to invade into Matrigel and adhere to ECM components are shown in Figure 5. Results of the wound-healing assay revealed that the rates of wound closure were similar for wt, control (sh-luc), and HSPA2-deficient (sh-3 and sh-4) cells (Figure 5a,e). In addition, no differences in capability of control and HSPA2-deficient NCI-H1299 and NCI-H23 cells for chemotactic migration toward the FBS gradient (1–10%) were observed using the Boyden chamber assay (Figure 5b,g). The results in Figure 5c revealed that the deficit in HSPA2 has no effect on the invasive ability of NCI-H1299 cells. Finally, using an ECM Cell Adhesion Array Kit, we found that *HSPA2* knockdown in NCI-H1299, NCI-H358, and NCI-H23 cell lines had insignificant effects on their ability to adhere to the ECM components (Figure 5d,f,h). Finally, we also search for potential correlation between HSPA2 expression and the resistance of HBEC and NSCLC cells to cisplatin (Appendix A). Our results showed that the sensitivity of the cells to the drug is not dependent on the protein level of HSPA2. Altogether, our results indicate that HSPA2 had a negligible impact on the migratory, invading, adhesive abilities, and chemoresistance of NSCLC cells. 

### 2.3. HSPA2 Knockdown Has No Impact on Growth, Migration, Invasion, and Adhesion to ECM Components of Breast (MCF7) and Cervical Cancer (HeLa) Cells

Having established that HSPA2 has a negligible role in supporting proliferation, colony-forming ability, and other representative features of malignant NSCLC cells, we decided to test whether this is also the case in widely used model cell lines of the breast (MCF7 cell line) and cervical cancer (HeLa cell line) [26,27,28]. First, we used two different *HSPA2*-targetting siRNA sequences (siRNA1 or siRNA2) to decrease *HSPA2* expression in MCF7 cells. We found that although the transfection of siRNA sequences into MCF7 cells effectively reduced the HSPA2 protein level (Appendix A), it had no impact on cell metabolic activity (Appendix A). Moreover, a 120 h long dynamic assessment revealed comparable growth rates for both the HSPA2-deficient and control cells (Appendix A). In addition, no differences in clone-forming ability were observed between control and HSPA-deficient cells (Appendix A). These results showed that siRNA-mediated silencing of *HSPA2* expression had no effect on the growth of MCF7 cells. 

The effectiveness of the stable *HSPA2* gene knockdown in MCF7 cells using shRNA sequences differed from minimal in the case of shRNAG3 (sh-G3 cells) to very high in case of shRNA4 (sh-4 cells) (Figure 1b and Figure 6a,e). The silencing was specific for the target gene; also, the levels of HSPA2 were similar in control sh-luc and wt cell lines (Figure 6a,e). We found that cells’ metabolic activity (Figure 6b,f), proliferation rate (Figure 6c,g), and colony-forming ability (Figure 6d,h) were not dependent on the levels of HSPA2 expression both in MCF7 and HeLa cells. 

Finally, we examined whether the stable knockdown of *HSPA2* expression would influence the migratory, invasive, and adhesive potential of MCF7 and HeLa cells. We used the same methodology as in the case of the NSCLC cells. Results of the wound-healing and Boyden chamber chemotactic migration assays revealed that the ability of HeLa cells to migrate from the wound edge toward empty space during 24 h and 48 h was independent of the HSPA2 protein level (Figure 7a). In addition, any HSPA2-dependent difference in the numbers of chemotactic cells migrating toward the FBS gradient both in the case of HeLa and MCF7 cells was visible (Figure 7b,e). Similarly, results of the transwell invasion assay showed that a comparable number of control and HSPA2-deficient HeLa and MCF7 cells were able to invade Matrigel (Figure 7c,f). Furthermore, we observed that a similar number of the control and HSPA2-deficient cells were bound to collagens (I, II, and IV), fibronectin, laminin, tenascin, and vitronectin (Figure 7d,g). HSPA2 depletion also did not affect sensitivity of MCF7 cells to cisplatin (Appendix A). These results indicated that similarly as in NSCLC cells, HSPA2 plays an insignificant role in supporting the migratory and adherent phenotype of breast and cervix cancer cells in vitro.

## 3. Discussion

It is clear that HSPA2 is critical for the development and function of male germ cells [4,19]. Several papers showed that HSPA2 has a universal contribution to cancer cell phenotype (review in [20]), but it is not expressed in corresponding non-tumorigenic cells [27,28]. In this study, for the first time, we directly compared the phenotypic effects of HSPA2 deficiency in cancer and corresponding normal cells. As a model, we selected non-tumorigenic bronchial epithelial and NSCLC cells. We revealed that the HSPA2 chaperone plays an essential role in epithelial cells, but its activity is non-essential in several NSCLC cell lines, as well as in MCF7 and HeLa cells that were shown by others to be HSPA2-dependent [26,27,28]. We found that *HSPA2* knockdown in the immortalized bronchial epithelial BEAS-2B cells impaired their colony-forming (Figure 2d) and adhesive abilities (Figure 4c), while it had no influence on the cell growth (Figure 2b,c) and migration (Figure 4a,b). On the contrary, we observed negligible effects of HSPA2 depletion on the growth (Figure 3, Figure 6, Appendix A), adhesion, migration, and invasion of NSCLC (Figure 5), as well as of breast and cervical cancer cells (Figure 7). Taking together our previous observations that HSPA2 deficit neither affects the growth nor chemoresistance of two NSCLC cell lines [18], our studies put the current belief on the universal cancer-related role of HSPA2 under question. This paper is the only one that exploited different methods for the inhibition of the HSPA2 gene expression (transient siRNA- or stable shRNA-mediated knockdown (using retroviral or lentiviral vectors; or CRISPR/Cas9 knockout) and used a validated anti-HSPA2 antibody with proven specificity [20]. Furthermore, we found that independently of the initial level of the endogenous HSPA2, the downregulation of its expression did not lead to growth inhibition. 

Previously, we found that human HSPA2 is highly expressed in basal cells in various multilayered epithelia and in bronchial epithelium, which is a type of pseudostratified epithelium [21]. So far, the understanding of HSPA2 significance in normal somatic cells is limited to epidermal keratinocytes [25]. It is noteworthy that epidermal keratinocytes [25] and BEAS-2B cells (this paper) showed similar phenotypic changes after *HSPA2* knockdown. Reduced clonogenic potential without changes in the cells’ proliferation rate seems paradoxical. However, the results of clonogenic assay not only provide information about cells’ ability to divide, but mainly report on their ability to sustain growth. This feature relies on effective cell adhesion and diminished dependence on external survival factors. Not all proliferating cells are capable to form colonies; moreover, colony-forming epithelial cells are heterogeneous in their capacity for sustained growth [29]. In the context of epithelial cells’ biology, the reduction of clonogenic potential and ECM adhesion is thought to pertain to cells that lost stem cell-like characteristics and entered a differentiation program [30]. Therefore, it can be speculated that the decrease in HSPA2 expression may promote the transition of epithelial cells toward a more differentiated phenotype. This shift might not include changes in migratory potential (of note, our unpublished results showed that HSPA2 knockdown did not affect keratinocytes’ motility either). In fact, we have found that HSPA2-deficient keratinocytes show accelerated differentiation in a reconstituted human epidermis model [25]. 

The human bronchial epithelium comprises basal cells, luminal ciliated cells, and mucin-producing goblet cells, so it is also possible that HSPA2 may selectively impact on some cell sub-population. The basal cells in the bronchial epithelium act as epithelial stem/progenitor cells to replenish lost specialized cells [31]. Thus, further studies are needed to disentangle whether and how spatiotemporal changes in HSPA2 expression may impact on the phenotype of a specific cell sub-population. The contribution of HSPA2 to the differentiation, morphogenesis, and homeostasis of a mucociliary airway epithelium should be further studied using a physiologically relevant three-dimensional co-culture system of HBEC at the air–liquid interface. Taking into account the universal presence of HSPA2 in the basal epithelial cells in various multilayered epithelia, it is tempting to suggest that HSPA2 could be a unique example of a differentiation-related chaperone. 

Our study extends the current knowledge on the roles of HSPA paralogs in the biology of bronchial epithelium. In this context, the available data deal mainly with the stress-inducible HSPA1, which is present in HBEC under physiological conditions. The activity of HSPA1 in the airway epithelium was linked to mucin secretion [32,33] and protection against genotoxic agents [34]. In turn, extracellularly-released HSPA1, as a damage-associated molecular pattern (DAMP) molecule, has been attributed to the inflammatory responses of the airway epithelium [35,36,37] and the pathogenesis of severe respiratory diseases [35,38]. However, the question of why HBEC, as well as other somatic cells, need HSPA2, HSPA1, and other cytoplasmic/nuclear HSPA paralogs remains unanswered. It is generally assumed that the activity of these chaperones is similar; hence, they are functionally redundant. Nevertheless, the results of several studies implicate functional diversity among cytoplasmic/nuclear HSPAs and suggest that they may interact with a different set of protein substrates or execute a distinct outcome for their client proteins [5,7]. We believe that multilayered epithelia constitute a physiologically relevant model that enables searching for potential non-overlapping functions of HSPA1 and HSPA2. An intriguing possibility is that HSPA1 may provide protection against cellular stress, while HSPA2 may represent a basal cell-specific chaperone involved in the regulation of homeostasis. In this context, it is worth noting that *HSPA2* knockdown in epidermal keratinocytes did not alter their sensitivity to heat shock, while it promoted transition of their phenotype toward a more differentiated state [25].

In this work, we showed that lung, breast, and cervical cancer cells are not dependent on HSPA2 in terms of their ability to proliferate, form colonies, migrate, invade, and adhere to the ECM components. These findings support and extend our previous observations that neither the selective depletion of *HSPA2* nor double knockdown of *HSPA2* and *HSPA1A/B* expression decreased the proliferation or viability of NSCLC cells (previously demonstrated in NCI-H1299 and NCI-H23 cell lines) [18]. Our results also comply with findings showing an insignificant impact of selective siRNA-mediated *HSPA2* knockdown on the viability of certain breast, colon, ovarian, and prostate cancer cell lines [17]. In addition, even the triple knockdown of *HSPA1A/B*, *HSPA5,* and *HSPA2* expression did not decrease the viability of breast cancer MDA-MB-468 cells [17]. Altogether, these results indicate a functional redundancy between HSPA2 and other HSPA paralogs in various types of cancer cells. A lack of correlation between HSPA2 expression and resistance of NSCLC and MCF7 cells to chemotherapeutic drug (Appendix A) conforms to findings in our previous study [18], showing that HSPA2 may not be involved in mechanisms of anticancer drug resistance. Observations that chemical pan-HSPA inhibitors effectively blocked the proliferation of NSCLC and breast cancer cells [18,39] further confirm that HSPA2 and other HSPAs are redundant in their functions. In turn, the viability of BEAS-2B cells was relatively insensitive both to pan-HSPA inhibition [18], and selective *HSPA2* knockdown (Figure 2b,c). Accordingly, our results conform to an accepted view that HSPAs are not critical to the viability of non-transformed cells. We postulate that these chaperones play a role in a stress-response mechanism and/or in supporting/controlling tissue homeostasis (e.g., cell differentiation).

Conversely to the results presented here and in other articles [18,39], some other studies pointed to HSPA2 as a promoter of malignant phenotype [26,27,28,40]. An attractive and acceptable explanation of these discrepancies is that the dependence of cancer cells on HSPA2 activity may arise only on the specific molecular background. Putative factors determining such dependence are unknown so far. However, it was suggested that the interplay of HSPAs machine with variable co-chaperones can provide functional diversity to the cellular roles of HSPA paralogs. Specifically, the J-domain protein (also referred to as DNAJ) and nuclear exchange factor (NEF) families have been suggested as factors that confer functional diversity to HSPA machines [41,42]. A recent report confirmed that distinct functions of HSPA1 and HSPA1L dependent on the intracellular co-chaperone background and specifically are directed by the differential interaction with preferred nuclear exchange factors (NEFs) [8]. 

Thus, in certain cancers, the availability of specific NEFs and other co-chaperones, and/or their post-translational modifications might regulate the requirement for HSPA2 input into the HSPA chaperone machine. Such a scenario could partially explain contrary findings on the significance of HSPA2 in various lines of NSCLC cells [18,40]. Nevertheless, that possibility cannot explain conflicting conclusions on HSPA2 role in supporting the viability of MCF7 and HeLa cells, as reported by our studies and the studies of Schlecht et al. [17] versus others [26,27,28]. Potentially, distinct phenotypes could be generated by transient and stable *HSPA2* knockdown. However, in our hands, both silencing strategies showed consistently that HSPA2 is dispensable for the growth of MCF7 cells (Figure 6 and Appendix A). 

Taking into account that complex mechanisms can control the impact of HSPA2 on the fate of cancer cells, it seems particularly important to evaluate the reliability and usefulness of HSPA2 as a cancer biomarker. So far, a high level of HSPA2 in tumor was reported as a negative prognostic factor in esophageal cancer [43], hepatocellular cancer [44], pancreatic cancer [45,46], and NSCLC [22]. In the case of breast cancers, conflicting results were published. Accordingly, immunohistochemistry-based studies linked a high level of HSPA2 in tumor with unfavorable clinicopathological characteristics and shorter overall as well as disease-free survival [27,47]. Instead, analysis of the transcriptomic data showed that a high level of *HSPA2* mRNA predicts a favorable outcome [48,49,50]. Thus, uncertainty as to the value of HSPA2 as a cancer biomarker underlines the necessity to use validated and reliable tools for HSPA2 detection in tumor samples. Bearing in mind that HSPA2 is a member of one of the most conservative protein families in nature, we recently tested the specificity of commercial anti-HSPA2 antibodies. Only one out of six analyzed antibodies (including those previously used in biomarker studies) ensured the specific detection of HSPA2 and did not cross-react with other HSPA paralog(s) [20]. Unfortunately, no information on potential cross-reactivity with other HSPA paralog(s) was provided for anti-HSPA2 antibodies used in a vast majority of functional studies on HSPA2 role in cancer cells (for details, we refer to [20]). 

The prevalence of cross-reacting antibodies in the market may cause the researcher to use RNAi-inducing particles that simultaneously inhibit the expression of other homologous genes in order to detect a decrease in the level of a studied protein. This eventually would result in an analysis of exaggerated phenotypes. Given that both pharmacological pan-HSPA inhibition [18,36,51] and dual knockdown of HSPA8 and HSPA1 [16,17] caused a strong anti-proliferative response in cancer cells, such a scenario seems possible. In fact, concerns on possible off-target effects in functional studies on HSPA1 were raised previously [52]. It was speculated that the strong anti-proliferative response of PC3 prostate cancer cells after antisense RNA particles-mediated HSPA1 knockdown [53] was due to non-specific and off-target effects; such an outcome was not generated using selective shRNA sequences for HSPA1 silencing [52]. 

Observations in this study also raise an intriguing and important question as to why a high level of *HSPA2* expression shows distinct prognostic significance in histologically different tumors [22,48]. One explanation of this surprising phenomenon is that HSPA2 can be the subject of various post-translational modifications that may differentiate its function or intracellular trafficking in cancer cells. It is possible that under some circumstances, HSPA2, as a signaling molecule, is released from certain types of cancer cells in order to modulate tumor microenvironment or immunological response. Such a scenario seems an attractive direction for future studies, bearing in mind that extracellularly released HSPAs, as DAMP molecules, are considered engaged in the regulation of immunological response [9,54,55].

## 4. Materials and Methods 

### 4.1. Cell Culture and Experimental Conditions 

BEAS-2B (virus-transformed non-tumorigenic bronchial epithelial cells; CRL-9609) and cancer cell lines: NCI-H1299 (non-small cell lung carcinoma, CRL-5803), NCI-H23 (adenocarcinoma, CRL-5800), NCI-H358 (bronchioalveolar carcinoma, CRL-5807), MCF7 (breast carcinoma, HTB-22), and HeLa (cervical carcinoma, CCL-2) were purchased from American Type Culture Collection (Manassas, VA, USA). Cells were cultured at 37 °C under standard conditions (5% CO_2_, 95% humidity, 21% O_2_ concentration). BEAS-2B cells were grown in serum-free bronchial epithelial growth medium (Lonza Ltd, Basel, Switzerland) and maintained at low density (less than 70% confluence). NSCLC cell lines were cultured in RPMI (Sigma-Aldrich, St. Louis, MO, USA), MCF7 in DMEM/F12 media (Sigma-Aldrich, St. Louis, MO, USA), and HeLa in RPMI/DMEM-HG (1:1 *v*/*v*), all growth media were supplemented with 10% heat-inactivated fetal bovine serum (EuRx, Gdańsk, Poland) and antibiotics (gentamycin or penicillin–streptomycin, all experiments were performed without antibiotics addition). Cells were regularly checked for mycoplasma contamination. 

### 4.2. Protein Extraction and Western Blot Analysis

Cells were seeded in 6-cm dishes with a maximum density of 50–70%. Total protein extracts were prepared as described previously [20]. Briefly, cells were lysed by scraping in RIPA buffer (1× PBS, 1% NP-40, 0.1% SDS, 0.5% SDC, 50 mM NaF, 1 mM PMSF) supplemented with protease inhibitor mixture. Total protein content was determined using the Protein Assay Kit (Bio-Rad; Hercules, CA, USA). Total proteins (25–35 μg) were fractionated by SDS-PAGE on 8% polyacrylamide gels and transferred on nitrocellulose membrane using the Trans Blot Turbo system (Bio-Rad, Hercules, CA, USA) for 10 min. Membrane was blocked in 5% nonfat milk/TTBS (0.25 M Tris–HCl (Ph 7.5), 0.15 M NaCl, and 0.1% Tween-20) and incubated with anti-HSPA primary antibody: anti-HSPA1 (1:5000 mouse monoclonal antibody, clone C92F3A-5, RRID AB_311860, Enzo, Life Sciences, Famingdale, NY, USA), anti-HSPA2 (1:5000 rabbit monoclonal antibody, clone EPR4596, RRID AB_10862351, Abcam, Cambridge, UK), anti-HSPA8 (1:7500 mouse monoclonal, clone B-6, RRID AB_627761, Santa Cruz Biotechnology Inc., Dallas, TX, USA), and horseradish peroxidase (HRP)-conjugated β-actin antibody (1:20000; clone AC15, RRID AB_262011, Merck KGaA, Darmstadt, Germany). The antibody–antigen binding was detected using HRP-conjugated secondary antibodies: goat anti-mouse (1:5000; AP124P, RRID AB_90456, Millipore Billerica, MA, USA), goat anti-rabbit (1:2000; AP132P, RRID AB_90264, Millipore Billerica, MA, USA), and visualized by Clarity ECL Western Blot Substrate (Bio-Rad; Hercules, CA, USA). Densitometric analysis of immunoblots was performed using ImageJ Software (National Institutes of Health, Bethesda, MD, USA) [56].

### 4.3. Generation of HSPA2-Deficient Cell Lines

#### 4.3.1. Generation of Lentivirally Modified Cell Lines 

The control non-targeting shRNA (shRNAluc, 5’-GTGCGTTGCTAGTACCAAC-3’) sequence and shRNA (shRNA3, 5′-GCCATCCTCATCGGCGACAAA-3′ [18,25]; shRNA4, 5′-GCGGCGAGAAGAACGTGCTCAT-3′ [18,25]; shRNAG3, 5′-CGGCGACAAATCAGAGAATGT-3′ [27,28]) sequences targeting coding sequence of the human *HSPA2* (Entrez Gene: 3306) gene were inserted into lentiviral vectors and exploited to generate stable HSPA2-deficient cell lines (NCI-H1299 and NCI-H23) as described earlier [18,25]. Other cell lines used in this study were generated as follows: cells were plated onto six-well plates (BEAS-2B, 1 × 10^5^/well; NCI-H358, MCF7, HeLa, 1.5 × 10^4^/well) and transduced with supernatants containing lentiviruses for 24 h at 37 °C with the addition of polybrene (BEAS-2B: 4 µg/mL, NCI-H358, and HeLa: 8 µg/mL, MCF7: 10 µg/mL). Stably transduced cells were selected in a standard culture medium containing puromycin (BEAS-2B, 1 µg/mL; NCI-H358, 1 µg/mL; HeLa, 2.5µg/mL; MCF7, 2 µg/mL). 

#### 4.3.2. Generation of CRISPR/Cas9-Modified Cell Lines

NCI-H1299 cells were seeded onto six-well plates (1.5 × 10^5^/well). The next day, cells were transfected with 1.5 µg of HSPA2 Double Nickase Plasmid (sc-400832-NIC, Santa Cruz Biotechnology, Inc., Dallas, TX, USA) or Control Double Nickase Plasmid (sc-437281, Santa Cruz Biotechnology) and Lipofectamine 2000 (Life Technologies, Carlsbad, CA, USA). After 72 h post transfection, cells were trypsinized and analyzed using the BD FACS Aria III Cell Sorter (BD Bioscience, San Jose, CA, USA). A blue laser was used for excitation (488 nm) and the fluorescein isothiocyanate (FITC) channel was used for green fluorescent protein (GFP) detection (530/30 nm). GFP-positive control cells (CRISPR-ctr) and knockout cells (CRISPR-A2) were sorted, seeded onto 6 cm plates, and used in further experiments.

### 4.4. Cell Proliferation, Metabolic Activity, and Clonogenic Assay

NCI-H1299 (2 × 10^3^/well), NCI-H358 (5 × 10^3^/well), MCF7 (6 × 10^3^/well), HeLa (3 × 10^3^/well), and BEAS-2B (4 × 10^3^/well) were seeded onto 96-well plates. At the indicated times after plating, the metabolic activity of cells was measured using CellTiter 96 Aqueous One Solution Assay according to the manufacturer’s protocol (Promega; Madison, WI, USA). The absorbance of the formazan product was measured (λ = 490 nm) using a microplate reader. The metabolic activity at the indicated time was calculated relative to the data obtained after 24 h of cell growth. For crystal violet proliferation assay, cells were seeded as described above, and at the indicated time after plating, cells were proceeded as described earlier [18,25], fixed with ice-cold methanol, stained with 0.1% crystal violet for 30 min, rinsed extensively with distilled water, and dried. Cell-associated dye was extracted with 10% acetic acid, aliquoted (100 μL), and the absorbance was measured at 595 nm using a microplate reader. To perform clonogenic assay, BEAS-2B (1.5 × 10^3^/well), NCI-H1299 (1 × 10^3^/well), NCI-H358 (3 × 10^3^/well), MCF7 (3 × 10^3^/well), and HeLa cells (1 × 10^3^/well) were plated onto six-well dishes, cultured for 7 (BEAS-2B, NCI-H1299, HeLa) days or 11 days (NCI-H358, MCF7) in growth medium, washed with PBS, fixed with methanol (−20 °C), dried, and dyed with crystal violet. Colonies were counted manually.

### 4.5. Adhesion Assay

Adhesion was analyzed using the ECM Cell Adhesion Array Kit (#ECM 540, Colorimetric; EMD Millipore, Billerica, MA, USA). Briefly, BEAS-2B (6 × 10^4^/well), NCI-H1299 (6 × 10^4^/well), NCI-H23 (7 × 10^4^/well), NCI-H358 (7 × 10^4^/well), HeLa (7 × 10^4^/well), and MCF7 (7 × 10^4^/well) cells were prepared as a single cell suspension in serum-free media. Cell suspension (200 μL) was added to the wells and assayed as per the instruction manual. Absorbance at 570 nm was measured using a microplate reader. 

### 4.6. In Vitro Wound-Healing Assay

BEAS-2B (4 × 10^5^/well), NCI-H358 (3 × 10^5^/well), and HeLa (3 × 10^5^ cells/well) cells were seeded into 24-well plates; NCI-H1299 (4 × 10^5^/well) were plated into 12-well plates to grow in a monolayer for 24  h under serum starvation conditions (1% FBS). Then, a sterile 20–200 μL pipette tip was held vertically to scratch a cross in each well. The detached cells were removed by washing with 500 μL PBS. Then, a pre-warmed culture medium was added again, and images of the scratched cells layer were captured. The scratch closure was monitored and imaged in indicated intervals using an inverted light microscope under a 10× objective lens.

### 4.7. Boyden Chamber Chemotaxis and Invasion Assays

#### 4.7.1. Chemotactic Migration

BEAS-2B (2 × 10^4^/chamber), NCI-H23 (5 × 10^4^/chamber), NCI-H1299 (5 × 10^4^/chamber), HeLa (5 × 10^4^/chamber), and MCF7 (8 × 10^4^/chamber) cells were trypsinized and resuspended in media supplemented with 1% FBS and plated in the wells containing insert chambers. Then, a 10% FBS-containing medium was added to the lower chamber to serve as a chemoattractant. Cells were allowed to migrate across the polycarbonate membrane with 8 μm pore size. Using a cotton swab, the non-migratory cells remaining in the upper chamber were removed, and the inserts were rinsed in PBS. The membranes were fixed with ice-cold methanol for 10 min and stained for 20 min with 0.2% crystal violet solution. Then, the inserts were rinsed multiple times with distilled water. The migratory cells were counted manually under an inverted light microscope using an average of six 10× fields per insert. In the case of BEAS-2B cells, the membrane was cut from the insert, the crystal violet dye was extracted with 10% acetic acid, and absorbance at 595 nm was measured using a microplate reader. 

#### 4.7.2. Invasion Assay

Cells were starved in a medium containing 1% FBS for 24 h before the experiment. Cells in serum-free DMEM medium were seeded into the upper chambers of each well (24-well insert, 8 μm pore size) coated earlier with 200 µg/mL Matrigel Growth Factor Reduced (Corning Incorporated, Corning, NY, USA). Following incubation of the gel layer, cells were plated at the same density as described in the chemotactic migration section. Medium containing 10% FBS was placed in the lower chambers as a chemoattractant. After the indicated time of incubation, cells on the upper membrane surface were wiped off, and the ones that invaded across the Matrigel-coated membrane were fixed with ice-cold methanol for 10 min and stained with 0.25% crystal violet solution for 20 min. The number of invasive cells was counted in six randomly chosen low-power fields for each membrane under an inverted light microscope (Zeiss, Germany).

### 4.8. Statistical Analysis 

Unless otherwise stated, all data were shown as mean ± standard deviation of the mean (SD). The difference significance between two groups was determined by the two-tailed *t*-test for independent samples. A *p*-value of less than 0.05 was considered statistically significant. Microsoft Excel was used for the analyses.

## 5. Conclusions

The data presented in this article clearly show that HSPA2 does not promote the malignant phenotype of NSCLC, breast, and cervical cancer cells. Our results contradict the current belief of the essential role of this chaperone in various types of cancer cells. Instead, we showed that HSPA2 expression is required by the non-transformed bronchial epithelial cells to maintain their clonogenic potential and adhesive features. Therefore, we are convinced that further research should concentrate on elucidating HSPA2 roles in epithelial cells. In relation to the cancer, it would be important to critically verify the alleged functional and clinical significance of HSPA2 in cancer cells using reliable, validated researched tools and experimental models.

## Figures and Tables

**Figure 1 cancers-12-02749-f001:**
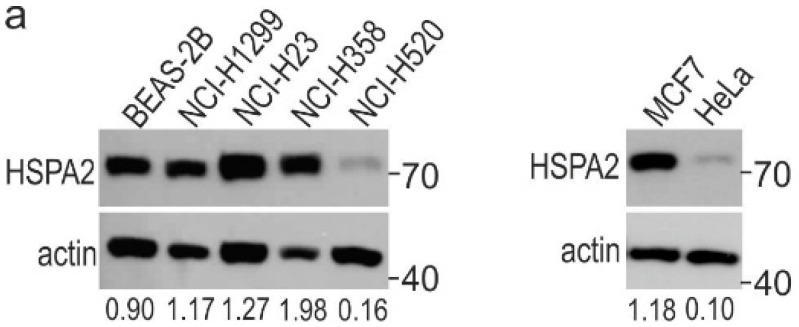
Heat Shock Protein A2 (HSPA2) level in wild-type cells (wt; black bars), modified control cells (sh-luc, CRISPR-ctr; gray bars), and HSPA2-deficient cells (sh-G3, sh-3, sh-4; yellow and green bars patterns, CRISPR-A2; green bar). sh-G3, sh-3, sh-4, and sh-luc cells were modified using the lentiviral vectors encoding the *HSPA2*-targetting and non-targeting shRNA sequences, respectively. CRISPR-ctr and CRISPR-A2 were generated using the CRISPR/Cas9 gene editing method. (**a**) Immunoblots showing HSPA2 level in wild- type cells. Representative immunoblots are shown (*n* = 3; molecular weight in kDa is indicated); actin was used as a protein loading control. Numbers below each lane represent the protein ratio normalized to the actin level. (**b**) Densitometric analysis of immunoblots (BEAS-2B, *n* = 4; NCI-H1299, *n* = 8; NCI-H23, *n* = 3; NCI-H358, *n* = 4; NCI-H520, *n* = 5; MCF7, *n* = 5; HeLa, *n* = 3) was performed using ImageJ Software. The relative protein level is shown after normalization to reporter protein level (actin). Statistical significance was calculated in relation to modified control using two-tailed *t-*test. CRISPR-A2, cells transfected with gene editing CRISPR/Cas9 (clustered regularly interspaced short palindromic repeats and CRISPR-associated protein 9) plasmids; CRISPR-ctr, control cells transfected with negative control plasmid. * *p* < 0.05 versus sh-luc or CRISPR-ctr cells, statistical significance was calculated using the two-tailed *t*-test.

**Figure 2 cancers-12-02749-f002:**
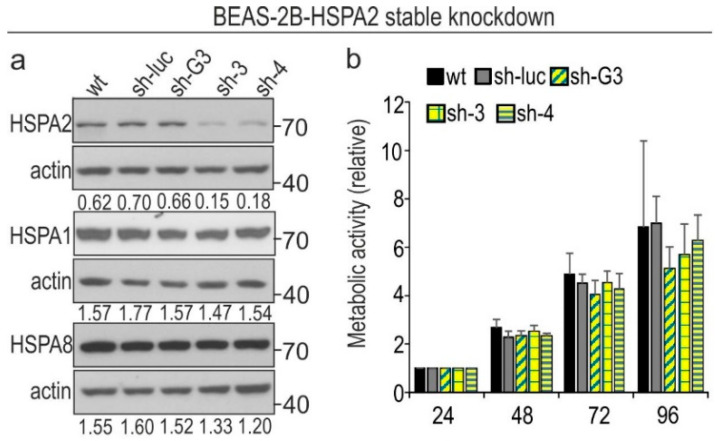
*HSPA2* silencing in the immortalized bronchial epithelial BEAS-2B cell line. (**a**) Immunoblots show the protein levels of HSPA2, HSPA1, and HSPA8 in wild type (wt); control sh-luc cells transduced with non-targeting shRNA-luc sequence; sh-G3, sh-3, and sh-4 cells transduced with HSPA2-targeting shRNAG3, shRNA3, or shRNA4 sequences, respectively. Numbers below each lane represent the protein ratio normalized to the actin level. Representative immunoblots are shown (*n* = 4; molecular weight in kDa is indicated); actin was used as a protein loading control. (**b**) Cell metabolic activity was tested using the MTS assay. Results are expressed as mean ± SD (*n* = 4, each in three technical repeats) in relation to values obtained at 24 h after plating. (**c**) Cell proliferation during 24–96 h of growth was examined using crystal violet staining assay (*n* = 4; each in three technical repeats). (**d**) Results of colony formation assay. Cells were plated onto six-well dishes (1.5 × 10^3^ cells/well) and cultured for 7 days. Colonies were counted manually (mean ± SD, *n* = 5, each in three technical repeats). Each dot color represents results of one of five independent experiments. * *p* < 0.05 versus sh-luc cells, statistical significance was calculated using the two-tailed *t*-test.

**Figure 3 cancers-12-02749-f003:**
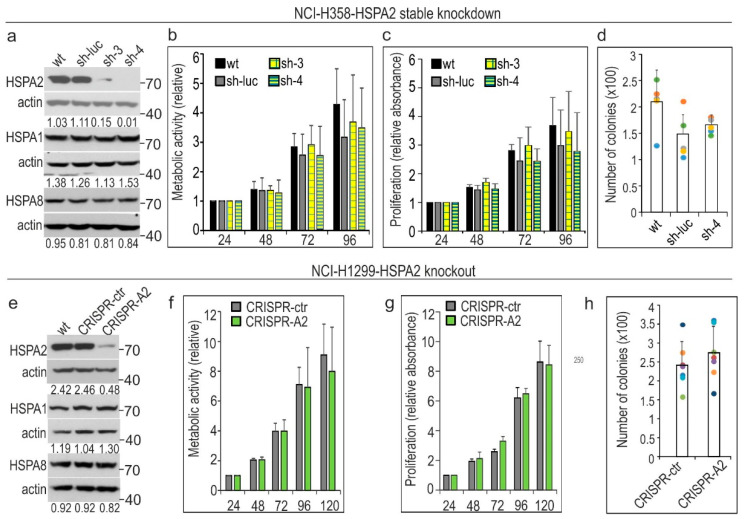
HSPA2 is dispensable for the growth of non-small cell lung carcinoma (NSCLC) cells. (**a–d**) Effects of stable *HSPA2* knockdown on the growth of NCI-H358 cells. Control sh-luc cells were stably transduced with the non-targeting shRNA-luc sequence; sh-3 and sh-4 cells were stably transduced with the *HSPA2*-targeting shRNA3 or shRNA4 sequences, respectively. (**e–h**) The *HSPA2* gene knockout in NCI-H1299 cells did not affect cell growth. CRISPR-A2, cells transfected with gene editing CRISPR/Cas9 plasmids; CRISPR-ctr, control cells transfected with negative control plasmid. Western blot analysis (**a**,**e**) of HSPA2, HSPA1, and HSPA8 levels in wild-type (wt) and modified cells. The numbers below each lane represent the protein ratio normalized to the actin level. Representative immunoblots are shown (*n* = 4; molecular weight in kDa is indicated), actin was used as a protein loading control. Cell metabolic activity (**b**,**f**) was measured using the MTS assay. Cell proliferation rate (**c**,**g**) was examined using crystal violet staining assay (*n* = 3; each in three technical replicas). Cell metabolic activity and cell proliferation rate are expressed as mean ± SD (*n* = 3, each in three technical replicas) in relation to values obtained at 24 h after plating. To analyze the clonogenic ability (**d**,**h**) cells were plated onto six-well dishes (1 × 10^3^ cells/well for NCI-H1299 and 3 × 10^3^ cells/well for NCI-H358 cell line) and cultured for 7 days (NCI-H1299) or 11 days (NCI-H358). Colonies were counted manually (mean ± SD, NCI-H1299 *n* = 7, each in three technical replicas; NCI-H358 *n* = 5, each in 2–3 technical replicas). Each dot color represents results of one of five (**d**) or seven (**h**) independent experiments.

**Figure 4 cancers-12-02749-f004:**
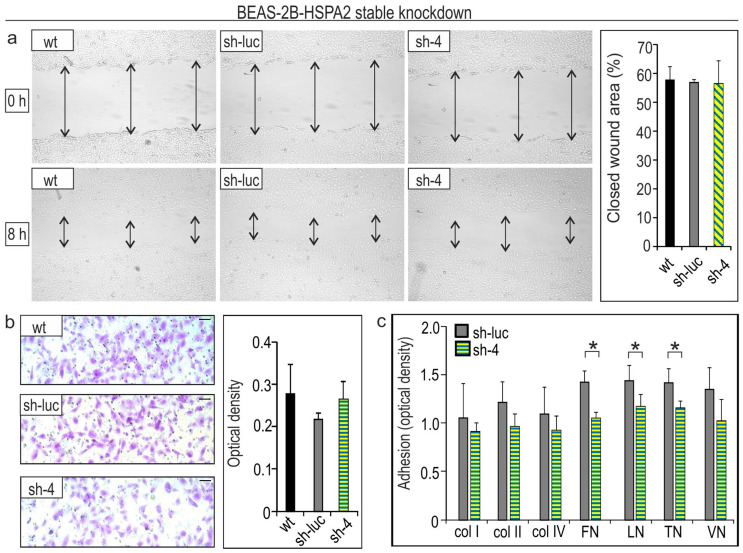
Effects of *HSPA2* knockdown on the ability of BEAS-2B cells to migrate and adhere to ECM components. (**a**) Cell motility assayed using the wound-healing assay. The open wound area was calculated at 8 h after making a scratch in confluent cell culture (**a**). Wild-type (wt); control sh-luc cells transduced with the non-targeting shRNA-luc sequence; sh-4 cells stably transduced with the *HSPA2*-targeting shRNA4 sequence. Results are shown as mean ± SD, *n* ≥ 4; each at least in eight technical repeats. (**b**) Cell chemotaxis toward fetal bovine serum (FBS) gradient (0.1–10%) assayed using a Boyden chamber system. Cells (2 × 10^4^) were plated onto an insert and allowed to migrate for 8 h. Representative photographs show cells that migrated through the membrane pores (8 µm). The scale bar represents 50 μm. The graph shows the optical density of cells (mean ± SD, *n* = 3, each in two technical repeats). (**c**) Cell adhesion to the extracellular matrix (ECM) proteins assayed using the ECM Cell Adhesion Array Kit (Millipore). Cells (6 × 10^4^) were plated onto the ECM Cell Adhesion Array strips and allowed to attach for 1 h (*n* = 2, each in two technical repeats). col, collagen; FN, fibronectin; LN, laminin; TN, tenascin; VN, vitronectin. * *p* < 0.05 versus sh-luc cells.

**Figure 5 cancers-12-02749-f005:**
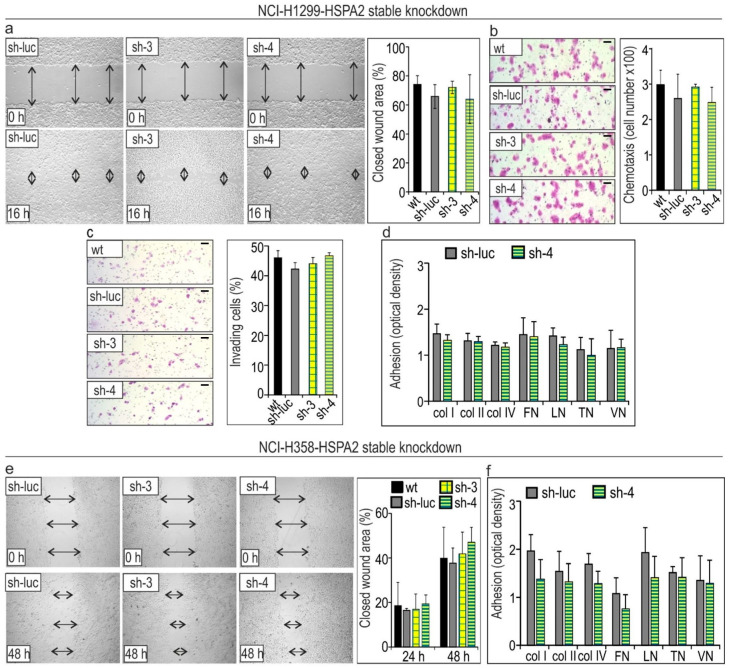
*HSPA2* knockdown has no effects on the motility, invasiveness, and adhesion of (**a**–**d**) NCI-H1299 cells; (**e**,**f**) NCI-H358 cells; and (**g**,**h**) NCI-H23 cells. (**a**,**e**) Results of the wound-healing assay. The microphotographs of a wound area were taken immediately after making a scratch in confluent cell culture (0 h) and after subsequent culture for 16 h (NCI-H1299) or 24–48 h (NCI-H358). Arrows mark the wound width. The graphs show mean ± SD (*n* = 4 in (**a**); *n* = 3 in (**e**); each at least in ten technical replicas). (**b**,**g**) Results of chemotactic migration assay. Cells (5 × 10^4^) were plated onto an insert and allowed to migrate for 6 h toward the FBS gradient (1–10%). Representative microphotographs show crystal violet-stained cells that migrated through the membrane pores. The graphs show mean ± SD (*n* = 5). (**c**) Results of cell invasion assay. Representative microphotographs show crystal violet-stained cells that migrated into Matrigel. The scale bar represents 50 μm. The graph shows the percent of invading cells (mean ± SD, *n* = 4, cells were counted in six different areas). (**d**,**f**,**h**) Results of adhesion assay. NCI-H1299 (6 × 10^4^/well), NCI-H358 (7 × 10^4^/well), and NCI-H23 (7 × 10^4^/well) cells were plated onto the ECM Cell Adhesion Array strips and allowed to attach for 1 to the ECM components: col, collagen; FN, fibronectin; LN, laminin; TN, tenascin; VN, vitronectin. Results show mean ± SD (*n* = 2, each in two technical replicas).

**Figure 6 cancers-12-02749-f006:**
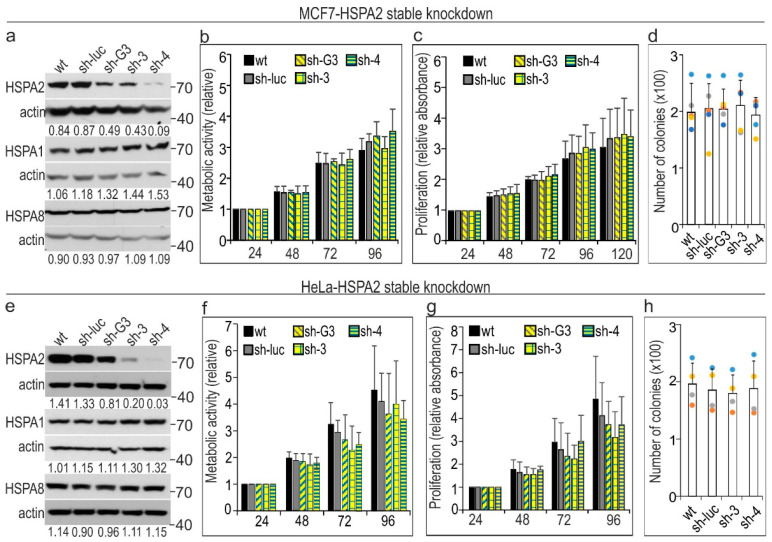
Stable shRNA-mediated *HSPA2* silencing in (**a**–**d**) breast luminal-A type carcinoma MCF7 cells and (**e**–**h**) human cervical adenocarcinoma HeLa cells. Immunoblots (**a**,**e**) show protein levels of HSPA2, HSPA1, and HSPA8 in wild type (wt); control sh-luc cells transduced with the non-targeting shRNA-luc sequence; sh-G3, sh-3, and sh-4 cells stably transduced with *HSPA2*-targeting shRNAG3, shRNA3, or shRNA4 sequences, respectively. The numbers below each lane represent the protein ratio normalized to the actin level. Representative immunoblots are shown (HeLa, *n* = 5; MCF7, *n* = 4), and actin was used as a protein loading control. Cell metabolic activity (**b**,**f**) was tested using the MTS assay. Results are expressed as mean ± SD (*n* = 5, each in three technical replicas) in relation to values obtained at 24 h after plating (**b**,**f**). The cell proliferation rate (**c**,**g**) was examined using crystal violet staining assay (*n* = 5; each in three technical replicas). For the clonogenic assay (**d**,**h**), cells were plated onto six-well dishes (HeLa 1 × 10^3^ cells/well, MCF7 3 × 10^3^ cells/well) and cultured for 7 days. Colonies were counted manually (mean ± SD, HeLa *n* = 4, each in three technical replicas, MCF7 *n* = 7, each in 2–3 technical replicas). Each dot color represents results of one of seven (**d**) or four (**h**) independent experiments.

**Figure 7 cancers-12-02749-f007:**
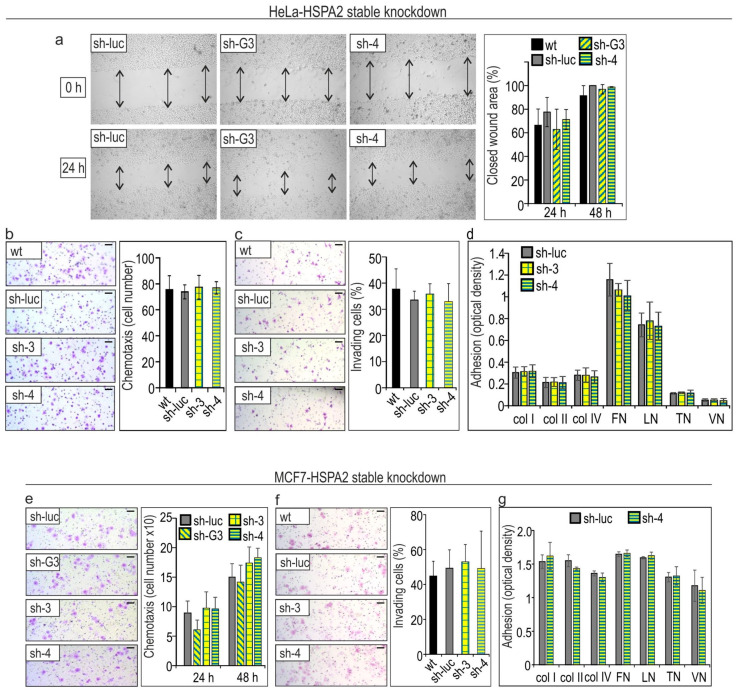
The motility, invasiveness, and adhesiveness of (**a**–**d**) HeLa and (**e**–**g**) MCF7 cells are not dependent on HSPA2 protein level. (**a**) Results of wound-healing assay. Representative microphotographs show the wound area immediately after making a scratch in confluent culture and also after 24 h culture. Arrows mark the width of wounds. The graphs show results (mean ± SD) of three independent experiments, each at least in ten technical replicas. (**b**,**e**) Results of a transwell migration assay. Cells (6 × 10^4^ (**b**), 7 × 10^4^ (**f**)) were plated onto the insert and allowed to migrate for 4 h (**b**) or 24–48 h (**f**) toward the FBS gradient (1–10%). Representative microphotographs show crystal violet-stained cells that migrated through the membrane pores. Graph shows relative chemotactic migration (mean ± SD, *n* = 3, each in two technical repeats). (**c,f**) Analysis of cell invasion into Matrigel. Representative microphotographs show cells that migrated into the ECM matrix. The scale bar represents 50 μm. The graph shows the percentage of invading cells (mean ± SD, *n* = 4, each time cells were counted in 6–7 different areas). (**d,g**) Cells (7 × 10^4^/well) were plated onto ECM Cell Adhesion Array strips and allowed to attach for 1 h. After staining with crystal violet, optical density was measured. Results show mean ± SD (*n* = 3).

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
