# Peer review of "HSPA2 Chaperone Contributes to the Maintenance of Epithelial Phenotype of Human Bronchial Epithelial Cells but Has Non-Essential Role in Supporting Malignant Features of Non-Small Cell Lung Carcinoma, MCF7, and HeLa Cancer Cells"

_cancers, 2020, doi:10.3390/cancers12102749_

Round 1

Reviewer 1 Report

The manuscript describes an in vitro study heavily limited by the use of specific cell lines that do not support general conclusions as the ones generally stated in the title and abstract about Non-Small Cell Lung Carcinoma, Breast and Cervical Cancer Cells. In this case the authors shoud use the name of the cell lines used.

An important limitation of this in vitro study is the lack of screening for expression levels of HSPA2 in different cancer cell lines, with a combination of testing in the cell lines showing various levels of endogenous HSPA2 expression. In those cases, the authors could also interfere with their shRNA and describe the outcomes.

Other specific comments or suggestions are:

-Since the clonogenic potential is sometimes hard to disentangle from the proliferation potential in vitro, the authors could use BrdU, instead of crystal violet, to better support their conclusion in Figure 1. A scatter plot or a box plot rather than the histogram in Figure 1D would help understand the extent of the observed difference. The same is asked for the next figures.

- It is quite surprising that the number of colonies in Figure 1D (non cancerous cells) is higher than the number of colonies in Figure 2D and H, showing the colony formation ability of cancer cells. Can the authors explain it?
Another comment to this regard is: The proliferation rate in Figure 1C is higher (for example at 72 hr) than those observed in Figure 2C and G. If this is consistent with reduced sensitivity in detecting colonies, can the authors argument on the biology of HSPA2 in proliferation and colony formation?

Reviewer 2 Report

Sojka and colleagues have tested the effect of HSPA2 knockdown on the behaviour of non-tumoral bronchial epithelial cells (BEAS-2B) and NSCLC cells in terms of growth, adhesion to ECM protein, and migration. The article is interesting in that it shows about the insensitiveness of NSCLC cells to HSPA2 knockdown as compared to BEAS-2B cells. I have some concerns, listed below, that affect the publication in its present form.

Major concerns

  1. HSPA2 knockdown in BEAS-2B cells impaired their colony forming (Figure 1d) and adhesive abilities (Figure 3d) while had no influence on the cell growth (Figure 1b,c), and migration (Figure 3a,b,c). Please explain why colony forming unit was hampered if cell growth and migration were not affected. Moreover, we need more mechanistic studies here, also because a similar trend was observed for keratinocytes (ref. 25). I suggest that a deep knowledge about this can be achieved with differentiation of BEAS-2B cells towards goblet/ciliated cells (see for example doi: 10.1155/2012/943982) in the presence of HSPA2 knockdown.  I say this also because some of the results in NSCLC lines have been already published by the same group. Thus, more the knowledge on BEAS-2B cells more the impact of this paper will be.
  2. CRISPR/Cas9 experiments shown in Figure 2 (e-h): although the author establish the rationale for using CRISPR/Cas9 to determine HSPA2 knockout, this task was not achieved, in that the HSPA2: actin ratio in these cells is 0.48 vs. 0.01 in lentivirus-mediated knockdown in previously Figure 2 (a). These results might be due to the use of two different cell lines (NCI-H1299 vs. NCI-H358). It is thus mandatory to perform the CRISPR/Cas9 experiments on NCI-H358 cells, also to have a direct comparison with two different methods of knocking down in the same cell line.
  3. A representation of the effect of HSPA2 knockdown on cell apoptosis should be shown.
  4. Why haptotaxis has been performed in BEAS-2B cells (Figure 3c) and not in the other cell lines? And viceversa, analysis of cell invasion was done with cancer lines but not with BEAS-B2 cells, why?

Minor concern

  1. In all Figures, the upper label concerning the cell’s names should be bound together: for example “BEAS-2B-HSPA2 stable knockdown”.
  2. In Figure 4, the chemotaxis results are expressed as “cell number/100”, whereas the same results of Figure 3 are shown as “optical density”. Please be consistent.
  3. In Figure 4, the wound experiments is represented as either “open wound area (100%)” or “closed wound area (100%)”, please be consistent.
  4. Figure 4b and Figure 6c,f: please replace in the y-axis “inviding” with “invading”.
  5. The assertion in the Legend to Figure 4 “Graphs (in b, d) show relative chemotactic migration (mean ± SD, n = 5).” is not true: b) represent chemotaxis, while d) cell adhesion. Please reformulate Legend to Figure 4 in its wholeness.
  6. In Figure 4g, the y-axis reports “cell number X10”, it is likely that is “cell numberX100”.

Reviewer 3 Report

The manuscript ID cancers-899297 by Sojka et al., aims to examine the phenotypic effect of HSPA2 chaperone using HSPA2 deficit in non-transformed human bronchial epithelial cells (HBEC), and in lung, breast and cervical cancer cells. This study was done comprehensively using proper control, methodology is explained clearly, however, study, neither bring any major new information nor able to clarify any existing doubt regarding the role of HSPA2 in cancer. In conclusion they mentioned that HSPA2 is not critically involved in proliferation and malignancy of breast and cervical cancer, and their data evidenced that, I wonder why they want a further investigation of for the role of HSPA2 in malignant phenotype of cancer cells. Authors need to address novelty issue in their manuscript.

Minor comments:

  1. Please provide western blot images with molecular mass markers. Several antibodies recognize bands which are not the proper molecular masses of the target protein, this issue is more prevalent with the use of different commercial antibody.
  2. The manuscript needs to go through spell check.

Round 2

Reviewer 1 Report

With the new supplementary figures, the revised manuscript appears more balanced, with improved precision in the results.

Major concerns were conceptually addressed in the rebuttal letter and partly (sufficiently) in the manuscript. The point of the interpretation of the clonogenic assays in association with HSPA2 functions remains experimentally unanswered. I agree when the authors say that "clonogenic test should not be regarded as a variant of proliferation assay", but no experimental evidence indicates here that HSPA2 is working on specific cell sub-populations, in vitro, (potentially) associated with a defined phenotype. In agreement with the fact that this point could be deeply addressed in another work, as it requires a bunch of experiments, my recommendation is to accept the manuscript in a revised version where the authors stress that additional experiments are required to disentagle the effect of the timing and dosage of HSPA2 on the phenotype of specific cell sub-populations.

Reviewer 2 Report

Please provide in the Discussion a comment about BEAS-2B cells on the colony forming unit assay and proliferation, as you detailed in the answer to my first major concern.
